# Chest X-Ray Pneumothorax Segmentation with the Multistep Post-Processing

**Author(s) names withheld**                                          EMAIL(S) WITHHELD

## Abstract

Nowdays automatic segmentation of the organs under the risk or even various diseases, including cancer lesions and other abnormalities, became very demanding. This significant growth can be partially explained by the recent achievements and performance quality of the Deep Learning approaches. One of the directions where accurate clinical diagnosis and computer-aided systems require such solutions is the problems which are very poorly visually distinguishable by human eyes. Current techniques from the computer vision and deep learning allow to solve complex problems with very high level of accuracy and simplify the clinical workflow.

In this work we present recent results of the pneumothorax segmentation from the chest X-ray images. Pneumothorax may appear in case of dull chest injury, as a continuation of hidden problems with the lungs, or even more there could be no reason at all for finding (Guptaa D., 2000). In several situations, lung collapse can turn out as serious threat to life.

We propose new method which includes the chest X-ray image segmentation pipeline with the multistep conditioned post-processing. As the result, we demonstrate significant improvement compare to any strong "baseline" by reduction of the pneumothorax collapse regions which are missed out and of false positive detections. Our results indicate very high accuracy and strong robustness of the algorithm confirmed by corresponding efficiency on the two stage test dataset with a priori unknown and absolutely different distribution. Final Dice scores 0.8821 and 0.8614 for "stage 1" and "stage 2" test sets respectively were resulted in top 0.01% standing of the private leaderboard on Kaggle competition platform.

**Keywords:** Neural Network, Deep Learning, Segmentation, Medical Imaging

## 1. Introduction

One of the tasks where deep learning approaches demonstrate its potential and strengths as robust is the medical image processing. In order to improve the detection quality and performance automatic AI-based solutions became very popular.

Pneumothorax segmentation is one of the complex problems to detect by human eyes, which can be solved with very high level of accuracy using the automated deep learning based systems and simplify the clinical workflow. In the usual workflow of the doctors pneumothorax is often detected by a radiologist and usually challenging to confirm, so there is a reason to have an precise AI-based algorithm to process these cases.

## 2. Pipeline Details

In this work we describe our method including ensemble of three LinkNet networks as the baseline (Chaurasia A., 2017) with various backbones: se-resnext50, se-resnext101 and SENet154. We trained all these pipelines for 40 epochs with Adam optimizer and CosineAnnealing scheduling, then 15 epochs with SGD optimizer and CyclicLR scheduling and 15 epochs again with initial setup. We applied this technique in order to get fast convergence and very precise fine-tuning on second and third training phases.

One of the difficulties was to control the strong imbalance in the dataset. We performed "non-empty" sampling for mini-batch selection to always keep positive samples inside. Then training was performed in 10-fold cross-validation split format.

One of the most essential steps in this pipeline refers to the post-processing step. In order to reduce the false positive detections, we used random search to define the binarization threshold and to select minimal surface area of small objects to remove. By this we determined the unique set of hyper parameters for all models based on the out-of-fold validation. Secondly, we applied dilation procedure to the averaged mask, but only in case of the cross-agreement within the test-time augmentation (TTA) outputs - we selected the level of confidence as 0.9 between all variations of predictions. Ensembling of all models was done not by the traditional averaging but via the union of the binary predictions according to the Dice agreement between at least two different models (Fig. 1). More detailed, we proposed to measure the accordance between the biniarized predictions from each model and then combine them via the union of the segmented regions if the accordance is above selected threshold.

## 3. Results

In order to properly train this pipeline and squeeze best performance we used the full-size 1024x1024 pixels images without any compression. One very essential remarks that there were no data pre-processing. Another significant point is that training was done with 10-fold cross-validation and batch size of 2. Training and evaluation of this method was done as part of the Kaggle pneumothorax competition on the publicly available dataset (kag) consisting of 10675 training images, 1372 "stage 1" test and 3205 "stage 2" test images with 21.5% average rate of positive pneumothorax samples.

Obtained results show very strong performance compare to other solutions - top 0.01% in the final leaderboard standing, confirmed by the solid accuracy metric such as Dice score of 0.8821 and 0.8614 for "stage 1" and "stage 2" relatively. By these evaluation it is noticable how the proposed approach is robust to the new and unseen data.

We also provide training and validation logs of 2 over 10 folds on Figure 2.

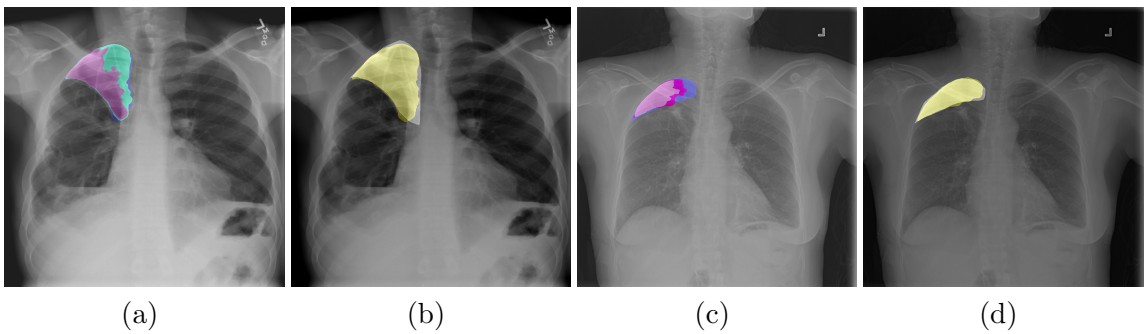

(a)            (b)            (c)            (d)

Figure 1: (a, c) - Comparison of mismatch between 3 different models predictions, (b, d) - Union ensembling [yellow] vs GT [gray]

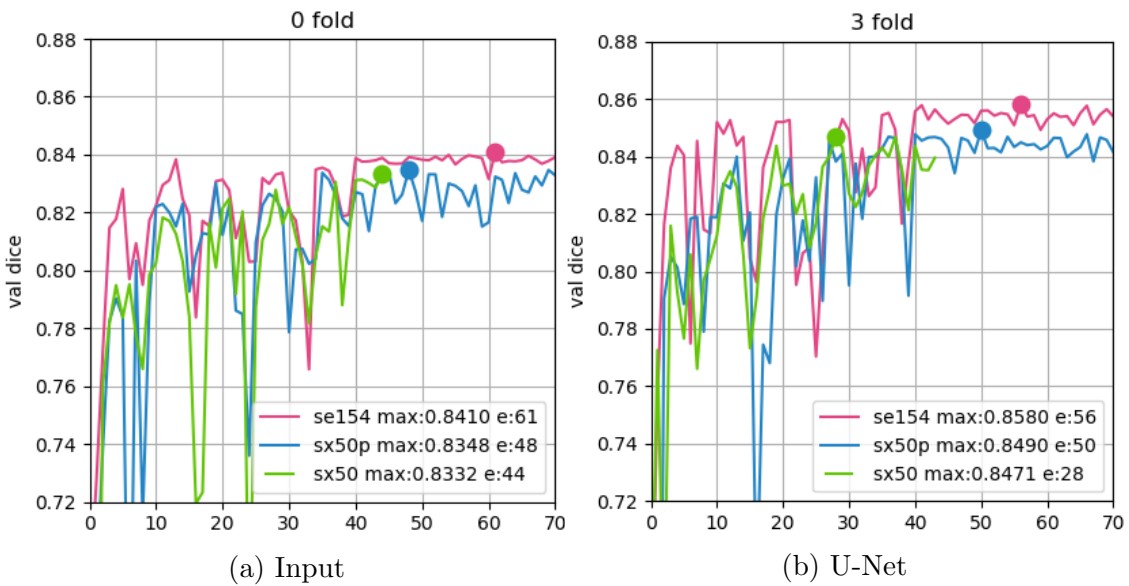

(a) Input            (b) U-Net

Figure 2: Logs of validation Dice scores of fold 0 and fold 3 though the epochs of training.

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
