# OpenReview forum: "Chest X-Ray Pneumothorax Segmentation with the Multistep Post-Processing"
_MIDL.io/2020/Conference — Submitted to MIDL 2020_

### Official Review · AnonReviewer3 · 2020-03-11
**Good results but very poorly written, with a unmotivated methodology**

**Rating:** 2
**Confidence:** 4

**Review:**

This paper is about pneumothorax segmentation on X-Rays, a medical condition that might be tricky to spot for the human eye.

The authors seem to reach very good performances (good ranking on the associated Kaggle competition), first by tuning very aggressively their hyperparameters (random search), plus adding miscellaneous heuristics as post-processing, with more hyperparameters.
The most interesting part is the ensembling performed, where they union the binary predictions of three models, as long as the models somewhat agree between them (one more hyperparameter to define that threshold).

To state things explicitly, I do not like methods with many hyperparameters that are tuned with heavy grid/random search. I think this is the opposite of machine learning, and it is just a way to overfit the validation set. There is no motivation on the choice of post-processing methods, so this looks like they mashed together different methods until it worked, without any intuition behind it.


The paper is very poorly written (plenty of typos, weird phrasing) and it is often difficult to make sense out of it. Few examples:

"Nowdays  automatic  segmentation  of  the  organs  under  the  risk  or  even  various  diseases,including cancer lesions and other abnormalities, became very demanding." -- Very demanding in terms of what ? Do you mean "in demand" ?

"Pneumothorax may appear in case of dull chest injury, as a continuation of hidden problems with the lungs, or even more there could be no reason at all for finding(Guptaa D., 2000). " -- I just cannot make sense out of it.


Based on the poor quality of the writing and methodology, I am choosing weak reject. Despite the good results, it doesn't seem to me that the authors would be able to communicate the method to other people, which is the point of a scientific conference (as opposed to a Kaggle leaderboard).

---

### Official Review · AnonReviewer4 · 2020-03-14
**Chest X-ray Pneumothorax segmentation**

**Rating:** 2
**Confidence:** 4

**Review:**

This manuscript describes a pipeline of methods for pneumothorax segmentation. The proposed method achieved very good results, ranking in the top 0.01% in the Kaggle competition. However, the method was not described clearly. It is hard to understand how each step is exactly applied to achieve good results. A flowchart would be helpful. Also more related work and ablation study would help show the literature and demonstrate the effectiveness of the work.

---

### Official Review · AnonReviewer2 · 2020-03-16
**Ensemble of CNNs with post-processing to segment pneumothorax on a Kaggle competition.**

**Rating:** 1
**Confidence:** 4

**Review:**

The paper proposes to train convolutional networks to segment pneumothorax by using three different parameters optimization strategies in cascade.
A number of subsequent post-processing techniques are used to refine the segmentation output, namely binarization, dilation, combination of binary masks across different models via union.
Training and validation is based on data from a public Kaggle competition.
The authors claim their results are within 0.01% in the leaderboard, but the reported performance could not be found in the current leaderboard (https://www.kaggle.com/c/siim-acr-pneumothorax-segmentation/leaderboard).
It is not clear what is the purpose of Figure 3, which shows performance curves for fold 0 and fold 3, and why the trends for these two specific folds are depicted here.

---

### Meta-Review · Area_Chair1 · 2020-04-07
**MetaReview of Paper15 by AreaChair1**

**Rating:** 1

**Metareview:**

There appears to be good results but the paper itself is not of adequate quality. Also issues with anonymity.

**Paper Type:**

both

---

### Decision · Program_Chairs · 2020-04-11

Reject